# Stratigraphic templates for ice core records of the past 1.5 million years

Eric W. Wolff[1], Hubertus Fischer[2], Tas van Ommen[3], David A. Hodell[1]

1. Dept of Earth Sciences, University of Cambridge, UK (ew428@cam.ac.uk)

2. Climate and Environmental Physics, Physics Institute & Oeschger Centre for Climate Change Research, University of Bern, Switzerland

3. Australian Antarctic Division and Australian Antarctic Program Partnership, University of Tasmania, Tasmania, Australia.

*Correspondence to:* Eric Wolff (ew428@cam.ac.uk)

**Abstract.** The international ice core community has a target to obtain continuous ice cores stretching back as far as 1.5 million years. This would provide vital data (including a $CO_2$ profile) allowing us to assess ideas about the cause of the Mid-Pleistocene Transition (MPT). The European Beyond EPICA project and the Australian Million Year Ice Core project each plan to drill such a core in the region known as Little Dome C. Dating the cores will be challenging, and one approach will be to match some of the records obtained with existing marine sediment datasets, informed by similarities in the existing 800 kyr period. Water isotopes in Antarctica have been shown to closely mirror deepwater temperature, estimated from Mg/Ca ratios of benthic foraminifera, in a marine core on the Chatham Rise near to New Zealand. The dust record in ice cores resembles very closely a South Atlantic marine record of iron accumulation rate. By assuming these relationships continue beyond 800 ka, our ice core record could be synchronised to dated marine sediments. This could be supplemented, and allow synchronisation at higher resolution, by the identification of rapid millennial scale-events that are observed both in Antarctic methane records and in emerging records of planktic oxygen isotopes and alkenone sea surface temperature (SST) from the Portuguese Margin. Although published data remain quite sparse, it should also be possible to match $^{10}Be$ from ice cores to records of geomagnetic palaeointensity and authigenic $^{10}Be/^9Be$ in marine sediments. However, there are a number of issues that have to be resolved before the ice core $^{10}Be$ record can be used. The approach of matching records to a template will be most successful if the new core is in stratigraphic order, but should also provide constraints on disordered records, if used in combination with absolute radiogenic ages.

## 1. Introduction

Ice cores have provided iconic records of changes in atmospheric composition and climate over glacial/interglacial cycles, with Antarctic datasets extending, so far, 800 kyr into the past (e.g. Bereiter et al., 2015; Jouzel et al., 2007; Wolff et al., 2010). While this illustrates what is often referred to as the "100 kyr world", marine (e.g. Lisiecki and Raymo, 2005) and terrestrial records indicate that a different style and strength of glacial cycle existed earlier in the Pleistocene, during the "41 kyr world". The change in amplitude and frequency is referred to as the mid-Pleistocene Transition (MPT) and occurs in the absence of any obvious change in astronomical forcing. The causes of the MPT remain hotly debated (Clark et al., 2006), with changes in $CO_2$ concentration or changes in the nature of the ice/rock interface underlying continental ice sheets often invoked.

Some of the issues surrounding these debates could be resolved if an ice core record, extending beyond the MPT and including records of past greenhouse gas concentrations, could be obtained. It has therefore become a key target of the ice core community to find a location to drill a core reaching as far back as 1.5 Ma (Fischer et al., 2013). Several projects to obtain such a core are partially underway, including the European Beyond EPICA project which plans to drill between 2021 and 2025 at a site known as Little Dome C (LDC). This site is only about 30 km from the site of the EPICA Dome C drilling (Fig. 1) that reached 800 ka, but is located on top of a subglacial highland, thus avoiding basal melting that led to loss of the oldest ice at Dome C. The Australian Million Year Ice Core (MYIC) project is targeting the same region of Antarctica.

A major challenge is to date such a core. Recently greenhouse gas concentrations were reported for ice as old as 2 Ma at a blue ice location of Allan Hills, Antarctica (Yan et al., 2019). While this provided tantalising snapshots of atmospheric composition, the dating, using the $^{40}Ar$ atmospheric increase method (see below), was too imprecise (with a quoted uncertainty of 110 kyr or 10% of age) to assign data unequivocally to particular parts of glacial cycles, or even to specific cycles. While this is a particular issue for discontinuous records such as those from blue ice, dating is also likely to be a major problem for a "standard" core, even assuming it is complete and continuous.

A number of methods can be used to try and date the ice older than 800 ka. As with the blue ice, absolute ages may be estimated from radiometric methods, including $^{81}Kr$ decay (Buizert et al., 2014; Crotti et al., 2021), and the growth in atmospheric concentration with time of $^{40}Ar$ (Bender et al., 2008; Yan et al., 2019), but both of these methods currently have large error bars at ages of 1 million years or more. The decay of cosmogenic isotopes (using the ratio of $^{10}Be/^{36}Cl$ to remove production rate variations) also has potential, but issues with $^{36}Cl$ loss at low accumulation rate sites (Delmas et al., 2004) have to be solved and the dating accuracy will likely be similar to the one using $^{81}Kr$ and $^{40}Ar$.

While absolute methods can indicate the approximate age of the ice within the pre-MPT period, the uncertainty is currently too large to answer many of the questions that are relevant to such ice. For example, if changes in $CO_2$ did occur, it will be important to determine exactly when they occurred, and at what rate. Did the changes occur only in particular parts of each glacial cycle? In what part of glacial cycles did millennial events still occur before the MPT? All these questions require an age scale that is precise to within at worst a few millennia.

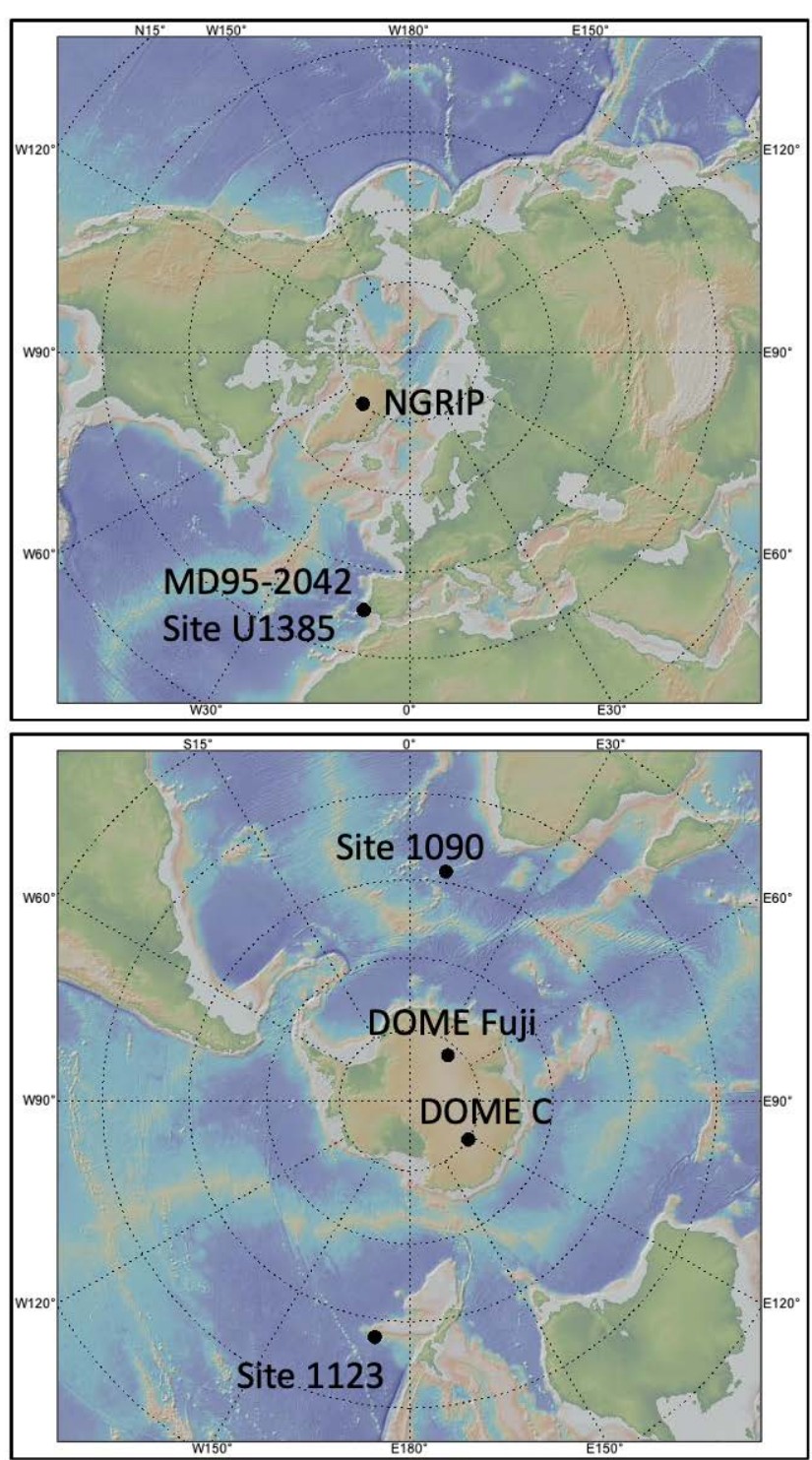

Figure 1. Location map. The maps show the locations of the ice and marine cores shown in this paper. Colours
represent topography and bathymetry. Figure made with GeoMapApp (www.geomapapp.org) / CC BY.
The main technique for dating the ice <800 ka in age has been to combine an estimate of past snow
accumulation rate and thinning with a range of fixed points that tie the ice core to known ages (Bazin et al.,
2013). These fixed points include the signal of low intensity of the geomagnetic dipole field associated with
polarity reversals recorded as increases in [10]Be deposition (Raisbeck et al., 2006), and various orbital tuning
targets including $\delta^{18}O_{atm}$ (Extier et al., 2018) and the ratio of $O_2/N_2$ (Kawamura et al., 2007). These methods, as
well as the radiometric ones, will certainly be applied to the new 1.5 Ma projects. However, diffusion, lack of
resolution and disturbed ice flow, with the possibility of folded ice near the bed, as has been seen at deep ice
core sites in Greenland (Grootes et al., 1993; NEEM Community Members, 2013), mean that further
stratigraphic methods to date the core may be needed.
One additional option is to create templates to which the records generated in the new projects can be matched.
The orbital targets (for tuning of $O_2/N_2$ and $\delta^{18}O_{atm}$) used to construct the 800 ka age model (Bazin et al., 2013)
are simple examples of the use of such templates, and will not be included here as they are very straightforward
to construct.  Marine and terrestrial records that are rather well-dated extend beyond 1.5 million years. As an
example many marine records have been mapped, using benthic isotopes, onto the LR04 marine stack (Lisiecki
and Raymo, 2007), whose age uncertainty at 1.5 Ma is estimated at 6 kyr, or the more recent Prob-Stack (Ahn et
al., 2017), which also uses the LR04 age model. Alignments such as these would allow age uncertainties of the
order needed to answer the questions about timing of $CO_2$ and millennial change discussed above. The
drawback is that they obviously preclude the option of assessing phasing between ice records and the (marine)
templates, as such a phase has to be assumed. In the absence of better dating methods, this cannot be avoided.
In this paper, we consider which ice core parameters may have analogues in the marine record that could be
used as templates onto which a future ice core could be mapped. We focus particularly on the EPICA Dome C
ice core (EDC), because its close proximity to the planned ice cores at LDC leads us to expect a similar signal in
most parameters. We use ice core datasets which have already been shown to closely mirror a particular marine
record over the past 800 kyr. We consider the mechanistic basis for such agreement and whether it is likely to
apply through the MPT to 1.5 Ma. We then present "predictions" of what some parameters might look like in
the new ice core, which can be used as both a test of integrity and continuity, and as a first dating tool for the
core.

## 2.   EPICA Dome C ice core records

In the following sections, we will consider possible analogues for 4 ice core parameters (Fig. 2). The water
isotope record ($\delta^{18}O$ and $\delta D$) is the most basic climate parameter (Jouzel et al., 2007) recorded in the ice,
generally considered to represent temperature at the ice core site. Dust is the insoluble component of impurities
trapped in the ice, and represents terrestrial material from the southern continents. Both dust and water isotopes
display particularly strong changes over glacial cycles with more subdued millennial scale variations.  Methane
is the one component in the ice core record that displays abrupt events, parallel to the rapid Dansgaard-Oeschger
events seen in Greenland ice cores. [10]Be (not shown in Fig. 2) is the cosmogenic isotope most commonly
measured in ice, and its production is controlled by changes in Earth's and the Sun's magnetic fields which also
influence cosmogenic isotopes archived in other material.  These 4 components will be considered in more
detail in the following sections.

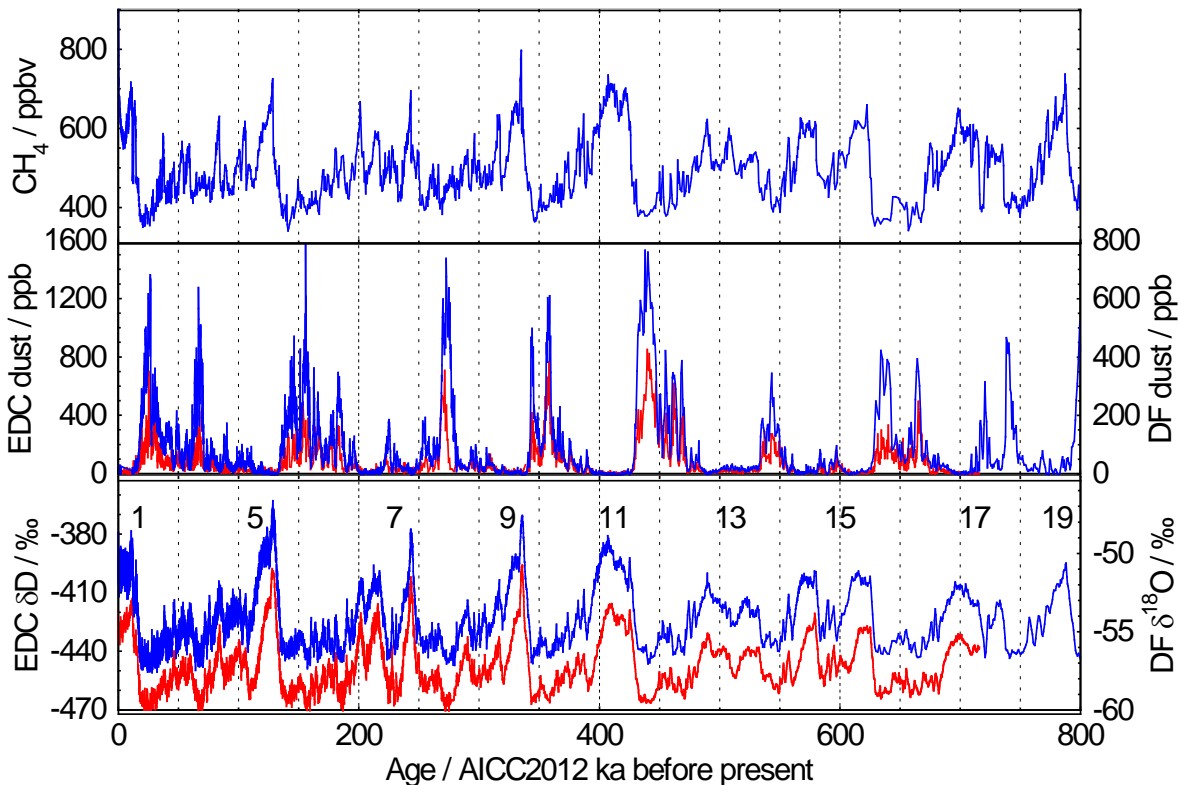


Figure 2. Ice core data over the past 800 kyr. Ice core records covering the past 800 kyr from Dome C (blue) and
from the last 720 kyr from Dome Fuji (red). Top panel: methane (Loulergue et al., 2008); middle panel: dust
(Kawamura et al., 2017; Lambert et al., 2008); lower panel: water isotopes (Jouzel et al., 2007; Kawamura et al.,
2017), with interglacial marine isotope stage numbers marked.

Although we are specifically aiming here to create a template for the European or Australian drilling at LDC, we
note that in most details, the features and relative changes seen for water isotopes, dust and [10]Be are expected to
be similar across the East Antarctic plateau. This is illustrated in Fig. 2, where we have plotted $\delta^{18}O$ and dust
concentration from Dome Fuji (Fig. 1) (Kawamura et al., 2017) along with $\delta D$ (Jouzel et al., 2007) and dust
concentration (Lambert et al., 2008) from EDC, all plotted on the AICC2012 age scale. The absolute level of
dust concentrations varies spatially across the Antarctic plateau, being dependent on travel distance from the
main Patagonian dust source region (Fischer et al., 2007a) and concentrations are higher at Dome Fuji than at
Dome C. This difference is explained in part because of the different analysis method used for Dome F and
Dome C that includes different size ranges, but in any case the pattern is almost identical  on multimillennial
timescales; methane is of course expected to show the same concentrations across Antarctica. Our templates for
LDC are therefore likely to serve as equally valid for other sites across East Antarctica.
**3.    Water isotopes**
Water isotopes ($\delta D$, $\delta^{18}O$) in ice cores are generally taken to represent the temperature at the ice core site,
although the reality is actually much more complicated than that (Buizert et al., 2021; Jouzel et al., 1997). It
therefore makes sense to look for a potential marine analogue that also records mainly temperature. Although
water isotope records from ice are sometimes plotted along with oxygen isotope records from marine cores, the

latter reflect a combination of temperature and ice volume (as well as local salinity effects), and so the variability of the two records may not be comparable on glacial-interglacial scales. A commonly used geochemical temperature sensor in marine cores is the ratio of Mg/Ca in foraminifera.

Planktic Mg/Ca records, covering 800 ka and more, are available from a number of marine sites and should reflect sea surface temperatures (SSTs) (e.g. Shakun et al., 2015). However while we expect some match between Antarctic temperatures and those from the high southern latitudes, we would expect most other sites to display a rather different pattern owing to the operation of the bipolar seesaw (e.g. Barker et al., 2011). Elderfield et al. (2012) noticed a striking similarity between the deepwater temperature inferred from Mg/Ca of benthic foraminifera at Ocean Drilling Program (ODP) site 1123 (Fig. 1), on the Chatham Rise east of New Zealand, and the temperature inferred from δD in the EDC ice core. They hypothesised that this is because deepwater temperature, particularly in the South Pacific, reflects the temperature of sinking surface waters and of Antarctic and proximal air temperature. Mean ocean temperature (determined by analysing noble gas ratios in ice cores) also shows a very similar pattern to Antarctic surface temperature across the last two glacial terminations (Baggenstos et al., 2019; Bereiter et al., 2018; Shackleton et al., 2020; Shackleton et al., 2021), which supports the interpretation. In recent years, deep water temperatures covering at least 1.5 Ma have been obtained from two other sites in the North Atlantic (Sosdian and Rosenthal, 2009) and North Pacific (Ford and Raymo, 2019), but given their location they are not expected to reflect Antarctic climate so directly, thus leaving only ODP site 1123 as a suitable comparator.

In Figure 3, we compare the record of δD from EDC with that of benthic Mg/Ca from site 1123 over the last 800 ka. The Mg/Ca record is presented as converted to temperature and interpolated (Elderfield et al., 2012), and is on the LR04 age model (Lisiecki and Raymo, 2005), while the ice core data is on AICC2012 (Bazin et al., 2013). By using a suitable amplitude scaling to overlay the two records we can compare their fidelity to each other and observe the extended Mg/Ca record as a possible template for δD over 1.5 Ma.

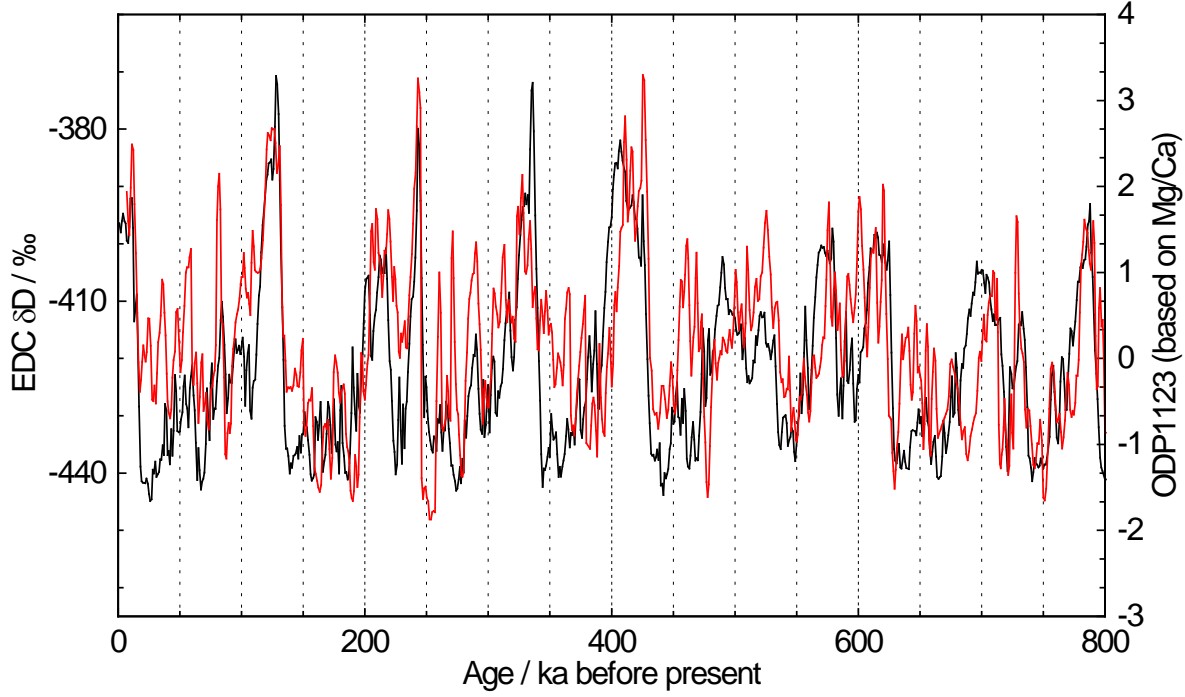

152

Figure 3. Ice core and marine sediment data reflecting temperature for the past 800 kyr. EDC deuterium (black, AICC2012 age scale) (Jouzel et al., 2007). ODP site 1123 deepwater temperature (red, LR04 age model), based on Mg/Ca (Elderfield et al., 2012).

The similarity between the two records is strong at the orbital timescale where both the shape and the relative amplitude of each glacial cycle is the same in the two records. The correlation coefficient between the two records after they are aligned in time is 0.67 (Elderfield et al., 2012) ($r^2 = 0.45$). However, there are significant mismatches at the shorter, multimillennial, timescale. Some very prominent millennial-scale AIM (Antarctic isotopic Maximum) events in the ice core record are very weak, or in some cases not clearly resolved, in the marine record. Some of the issues may actually be related to temporal synchronisation, and perhaps to the resolution of the marine record.  But still, it would be hard to use the marine record as a template for an ice core record between 450 and 550 ka (MIS 13). This is a concern because some of the sections of Site 1123 beyond 800 ka have a similar nature (in terms of signal amplitude) to that section.

Despite these concerns over the fidelity of the marine record as a predictor of the ice core isotope signal, we would expect the similarity to continue provided deepwater temperature at high southern latitudes continues to be driven by surface temperatures around Antarctica before the MPT.   What could disrupt such a link would be significant changes in ocean circulation and in the reach of different water masses. Such changes may well have occurred over the MPT (Ford and Raymo, 2019), and one suggestion is that they are related to a hypothesised change in the Antarctic Ice Sheet (Raymo et al., 2006) from largely terrestrial to marine-based.  While such changes would certainly have impacted the supply of water affected by Antarctic surface temperatures to the deep ocean, the proximity of site 1123 to Antarctica makes it unlikely that a southern influence was completely absent at that time. We therefore see it as likely that the site 1123 Mg/Ca record extended to 1.5 Ma (Fig. 4) does serve as an approximate template for at least the glacial/interglacial variability in Antarctic temperature and

therefore LDC deuterium. However, we accept the possibility that the exact nature of the relationship between
the two records could have differed in the early part of the period from that observed after 800 ka, and indeed
should a mismatch be found in the ice core record it will provoke reconsideration of the assumptions made here.

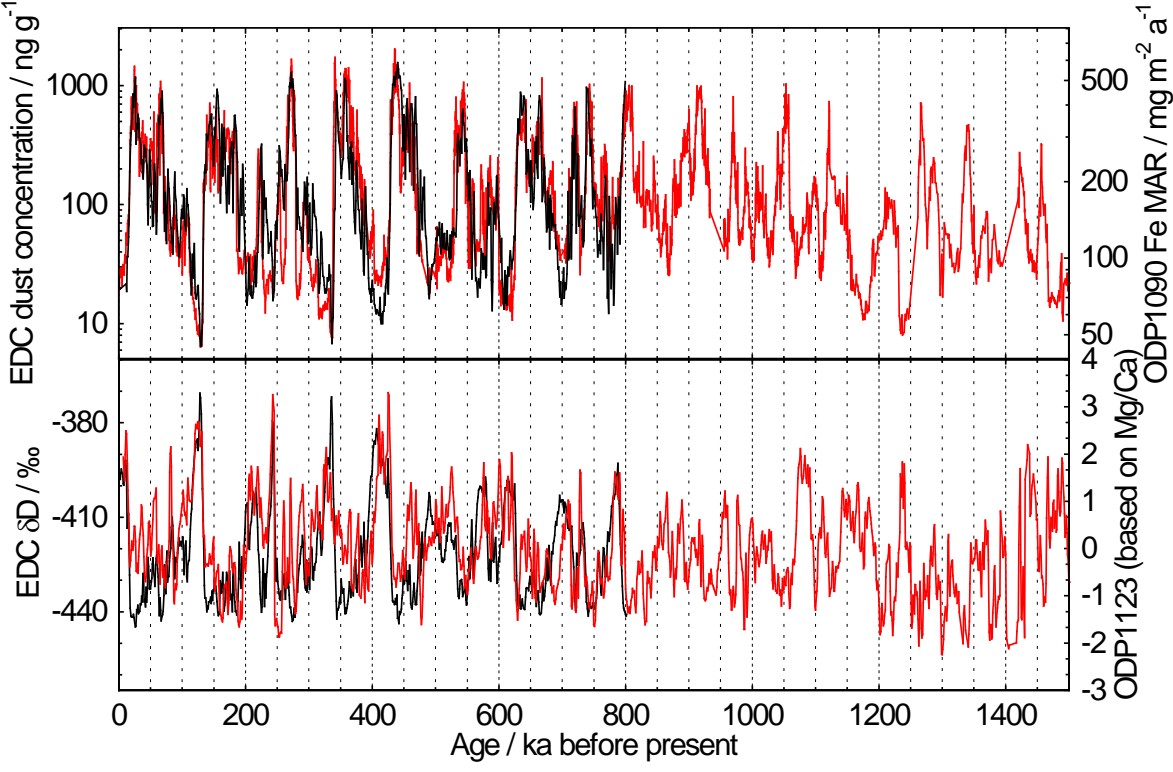


Figure 4. Ice core records to 800 ka and marine records to 1500 ka. Lower panel: EDC deuterium (black)
(Jouzel et al., 2007) and Mg/Ca-based deepwater temperature (red) from site ODP1123 (Elderfield et al., 2012).
Upper panel: EDC dust (black) (Lambert et al., 2008) and Fe MAR from ODP site 1090 (red) (Martinez-Garcia
et al., 2011).

It would obviously be beneficial to search for other marine analogues of Antarctic temperature. The similarity of
benthic oxygen isotopes in cores on the Portuguese Margin to Antarctic temperature was noted previously,
albeit on a very short time period (Shackleton et al., 2000). The extension of this record, which is underway
(Birner et al., 2016) would provide a much better resolved record, with clear millennial scale signals, and its
applicability as a template could be assessed. The caveat is that the underpinning reason for similarity of a
benthic isotope record controlled by several factors (ice volume, temperature, hydrography and water mass
changes) with Antarctic temperature at such a distal site is unclear, making it difficult to assess the likelihood
that the relationship persisted before the MPT.
**4.  Dust**
Terrestrial dust is measured directly in ice cores, as insoluble particle numbers and sizes which can be converted
to mass concentrations, and indirectly in marine sediments through the concentrations or ratios of elements that
are mainly (at appropriate sites) of windborne terrestrial origin. The 800 ka record of dust concentration in the
EDC ice core (Lambert et al., 2008) shows strong glacial-interglacial cycles, with high concentrations of dust in
glacial periods, and some multimillennial scale variability.  Elemental and isotopic analysis indicates that the
dust mainly originates from South American sources, particularly in Patagonia (Delmonte et al., 2008). As a
result we would expect a close relationship between dust arriving in Antarctica and dust deposited onto the
South Atlantic during the early stages of the path to Antarctica.
ODP site 1090 (Fig. 1) is ideally located to sample dust during its transport in the westerly wind belt from the
Patagonian sources towards Antarctica. Martinez-Garcia et al. (2011) noted that different dust proxies in the
sediment core from site 1090 matched well with each other over 4 Ma, and with EDC dust flux over 800 ka.
Here we compare their preferred dust proxy (mass accumulation rate of iron, Fe MAR) with the dust
concentration at EDC (Lambert et al., 2008). We prefer to use concentration rather than flux because this is what
we will be able to measure in the deeper parts of the LDC core – the flux is a derived quantity that requires
knowledge of the snow accumulation rate. It is therefore a fairer test to assess the similarity of the measured
quantity (concentration) to the marine target. In the section of the paper on [10]Be, we do discuss the potential to
use water isotopes to derive snow accumulation rate and hence calculate fluxes. This could also be done for
dust, if it were considered essential, but it carries some risk because of the assumptions involved.
In Fig. 5 we compare the two records over the last 800 ka, showing the result both for EDC dust concentration
and for flux. Note that glacials have high values of dust, that both records have been smoothed to 1 ka averages,
and are plotted on log scales.  With the appropriate amplitude scaling, the agreement between the two records is
remarkable.  This applies both to the consistent amplitude relationship, to the shape of glacial-interglacial
cycles, and to the identification of almost every multimillennial scale peak in both records. The section from
450-550 ka, which was problematical in the isotope records discussed above, shows a good match.   The
correlation coefficient between EDC dust and ODP1090 Fe Mar is 0.83 ($r^2$=0.69) both for EDC dust
concentration and flux.

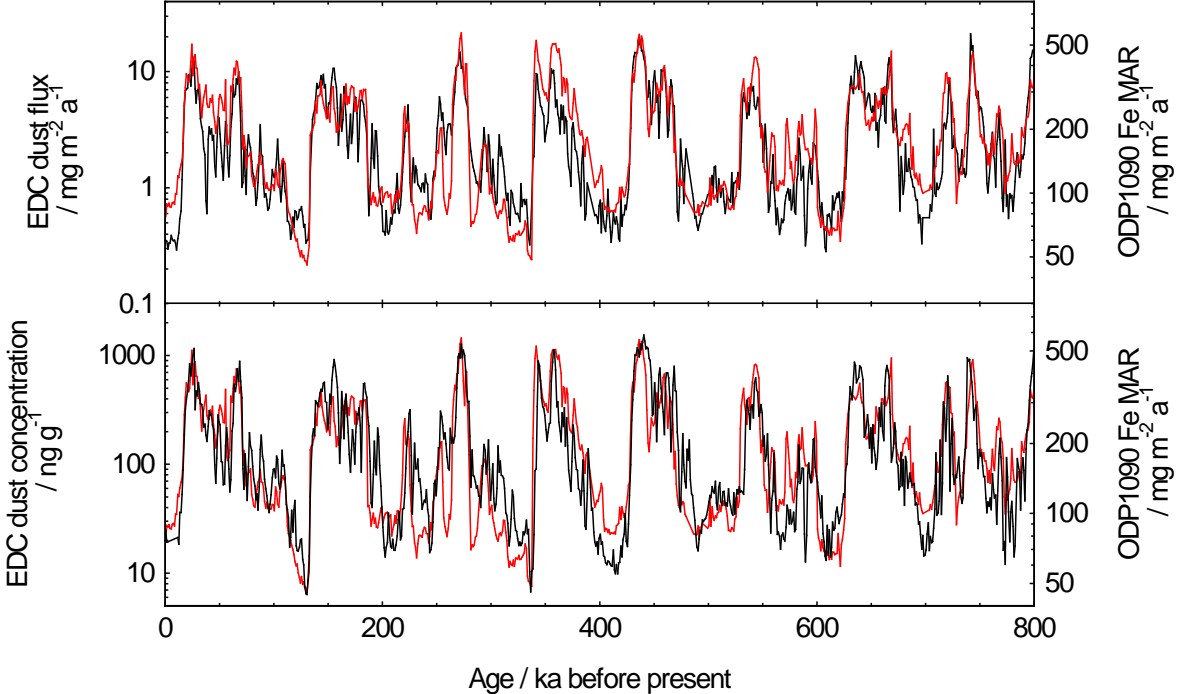


Figure 5. Ice core and marine sediment dust data for the past 800 kyr. EDC dust (black) (Lambert et al., 2008);
Fe MAR from ODP site 1090 (red) (Martinez-Garcia et al., 2011). Top panel uses EDC dust flux, while lower
panel uses EDC dust concentration. Data have been smoothed to 1 kyr averages and the marine data were
aligned (Martinez-Garcia et al., 2011) to the ice core age model.

The agreement is made more surprising by the fact that the dynamic range of the two records is very different:
the marine dust, being geographically closer to the dust production in Patagonia than is the long-range
transported ice core record, varies by a factor 10 (minimum in MIS 5e, maximum in MIS 13), while the ice core
dust concentration varies by a factor >100 (factor 200 between MIS 5e and MIS 13).   The range of dust flux at
EDC would be about a factor 50, because the snow accumulation rate is about four times higher in MIS 5e than
in MIS 13. This implies that the causes of dust variability are split into two halves: a factor of about 10 is due
mainly to changes at or near the source of the dust, another factor of about 5 is due to changes in lifetime during
the long meridional journey to Antarctica. This has been discussed several times before (Fischer et al., 2007b;
Lambert et al., 2008; Markle et al., 2018; Petit and Delmonte, 2009; Wolff et al., 2010) and although the
different approaches led to somewhat different amplification factors by dust source and transport processes, the
comparison shows that solutions that match the available data must consider changes both in source and in
lifetime.  We note that it is the very high dynamic range of the dust concentration or flux record in ice that
makes it possible to use raw concentrations in our comparisons – the relatively small factor change in
accumulation rate is overwhelmed by the factor 50 flux change, so that the same features seen in marine and ice
dust flux are still seen in ice concentration.
This implies that the extended marine dust record (Fig. 4) could be an excellent template for the dust record
expected in the LDC ice cores.  The part of the variance that is based on changes at or near the source should
remain, whatever occurred across the MPT. The second part of the variability, arising from changes in aerosol
lifetime over the Southern Ocean, has been in phase with changes at the source over the last 800 ka. This could
in theory have altered if there were major changes in atmospheric circulation across the MPT. Nonetheless, the
major part of the variability (that arising from changes at the source and at the start of the transport route) will
have remained unchanged. This implies that the basic glacial-interglacial pattern, as well as the imprint of
millennial scale change will have persisted. The one circumstance in which this would not be the case is that in
which a substantial new, local source of dust from the Antarctic margins existed. It has been suggested that one
aspect of the MPT might have been a change in the Antarctic ice sheet, with more terrestrial (rather than marine)
margins before the MPT (Raymo et al., 2006). Before using the dust record in the way we have proposed, it will
be important to check for the presence of new dust sources, through for example isotopic analysis of dust to
fingerprint its source area (Delmonte et al., 2008).
**5.    Methane as a pattern for Dansgaard-Oeschger variability**
While the EDC water isotope and dust records show strong variability, particularly on orbital timescales, that
can be used for pattern matching, their variations tend to be smooth, so that correlation is clear but imprecise.
Records containing the imprint of Dansgaard-Oeschger (D-O) events have the capacity to identify sharp time
points, and therefore to give much closer synchronisation, and many more clear tie points. Using the model of
the bipolar seesaw, it is possible to rather convincingly reproduce D-O events from the Antarctic isotope record,
to produce what is known as the synthetic Greenland record ($GL_T$-syn) (Barker et al., 2011). However, the
synthetic record can never have the sharpness of the original signal and in particular for ice older than 800 kyr
diffusion in the ice may have smoothed the higher frequency climate signal in the water isotope record. The only
record in Antarctic ice that does retain the character of the D-O events is the methane record.
Over the last glacial cycle, every significant D-O event recorded in the Greenland ice core record (North
Greenland Ice-Core Project (NorthGRIP) Members, 2004) is also seen in the EDC methane record (Loulergue et
al., 2008) (Fig. 6). The same pattern of abrupt climate change is seen in many other northern hemisphere
climate records, with a particularly faithful representation observed in planktonic oxygen isotope and alkenone
SST data from marine sediment cores from the Portuguese Margin (Govin et al., 2014; Shackleton et al., 2000).
Note that the benthic $\delta^{18}O$ from the Portuguese Margin strongly resembles the water isotope record from
Antarctica (here, $\delta D$ from EDC) and that the phasing of planktic and benthic $\delta^{18}O$ on the Iberian Margin is the
same as that seen between $CH_4$ and $\delta D$ in the Antarctic ice core record. This pattern has been interpreted as
being indicative of a thermal bipolar seesaw, and offers another signature for matching ice core and marine
records.
While the planktonic oxygen isotope and alkenone SST records reproduce the NGRIP (Greenland) ice core
isotopic record well in terms both of shape and amplitude, the methane record is less easy to match to the marine
record. This is because the amplitude of methane change in comparison to isotopic change (in either the
Greenland ice or North Atlantic marine record) is very variable. For example Greenland Interstadial (GI) 19 (at
73 ka) is very strong in the isotope records but shows a methane amplitude of only 70 ppb (Baumgartner et al.,
2014), and a sensitivity (methane jump/Greenland temperature change) less than a third of some other events.
This means that, in an unknown section of older core, we could only expect to make unequivocal matches for
some D-O events.

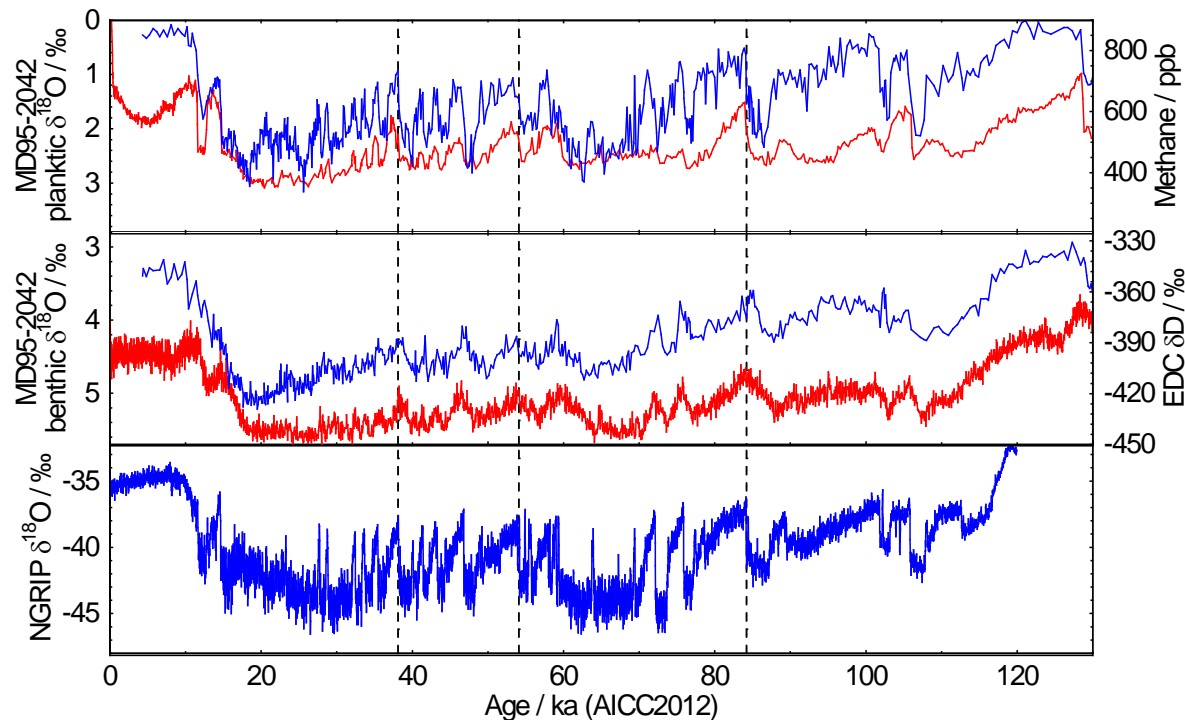

Figure 6. Sharp millennial scale features in the last glacial cycle. Bottom panel: NGRIP δ18O showing the pattern of Dansgaard-Oeschger events (North Greenland Ice Core Project Members, 2004). Middle panel: Benthic δ18O (blue) from site MD95-2042 (Govin et al., 2014; Shackleton et al., 2000) and the deuterium record from the Antarctic EDC ice core (Jouzel et al., 2007). Upper panel: Planktonic δ18O (blue) from site MD95-2042 (Govin et al., 2014; Shackleton et al., 2000) showing the same pattern as Greenland δ18O; methane from the Antarctic EDC ice core (red) (Loulergue et al., 2008) showing a more subdued version of the same variability. Vertical lines mark examples of the sharp onsets of three interglacials (Greenland Interstadial (GI) 8, 14 and 21).

High resolution isotopic data and alkenone SST collected at site U1385 (Fig. 1), extending to 1.45 Ma (Hodell et al., 2015), located close (25 km) to core MD95-2042 (discussed above and shown in Fig. 6) indicate that events of a D-O nature extend throughout the past 1.45 Myr (Birner et al., 2016). Thus the planktonic isotope and SST records from that site, soon to be published, should serve as a regional template for D-O variability. Using it with the methane ice core record makes the assumption that the teleconnection between North Atlantic climate variability and the (predominantly tropical) methane sources (Bock et al., 2017) remained intact before the MPT. This could be tested if East Asian speleothem records extended deeper in time than is currently the case (Cheng et al., 2016).

As an example of the potential for this method, we examine the relationship between the oldest part of the EDC record (MIS 19) and the equivalent data from site U1385 (Fig. 7). Here we can clearly identify the three strong millennial events on the MIS 19/18 boundary in both the marine and ice core record, with the sharp onsets in planktonic δ18O (marine) and methane (ice) and the more symmetric change in benthic δ18O (marine) and δD (ice). Carbon cycle data in both records also show the signature of the events. Radiometric ages are also

available in records that show these millennial events, adding further value to the correlations (Giaccio et al.,
2015). We note that one event (at about 780 ka) in the ice core record is not observed in the marine record,
despite the record having adequate resolution for its appearance.

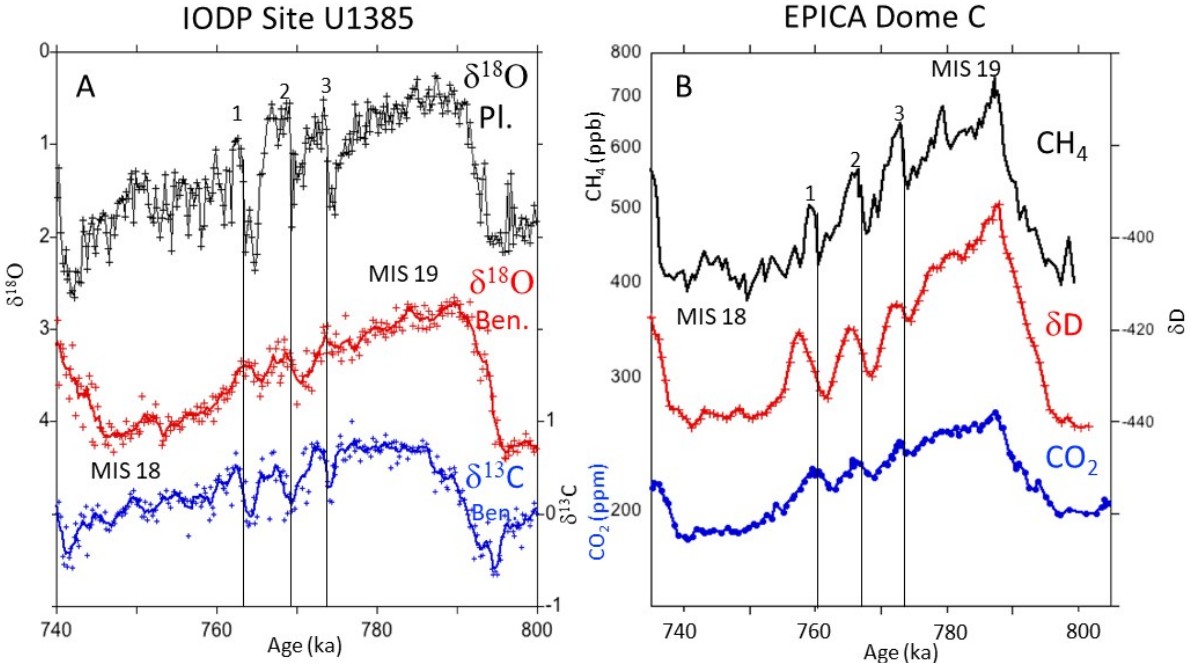


Figure 7. (A) Planktonic $\delta^{18}O$ (black), benthic $\delta^{18}O$ (red) and $\delta^{13}C$ (blue) over the MIS19-18 transition at Site
U1385 (Sánchez Goñi et al., 2016) compared to (B) $CH_4$ (black), $\delta D$ (red) and atmospheric $CO_2$ (blue) in the
EPICA Dome C ice core. Three strong millennial events (labeled 1-3) occur on the MIS19-18 transition that are
recorded in both the marine sediment and ice cores. Vertical dashed lines are drawn at the abrupt transitions
from cold stadials to warmer interstadial conditions. Note that the phasing of ice core $CH_4$ and $\delta D$ is not quite as
expected from later time periods (Fig. 6) and may reflect uncertainty in $\Delta$-age (the age difference between the
ice and gas records).

The variable amplitude of methane peaks relative to North Atlantic records may make it harder to use than some
other records. Nonetheless the simplicity of the match at ~770 ka suggests that methane in ice for the most
prominent millennial-scale features, used in a complementary way with other records, and matched against the
Portuguese Margin datasets, will provide a viable way of aligning the marine and ice records rather precisely at
least for the cycles immediately below 800 ka.  In the highly thinned ice over 1.2 Ma old, where there may be
>10 ka/m of ice (Lilien et al., 2021), the use of high-resolution continuous online laser spectrometric
measurement techniques to measure $CH_4$ (Chappellaz et al., 2013; Rhodes et al., 2015) should still allow
resolution of millennial features provided diffusion of methane (Bereiter et al., 2014) and of $\delta D$ (Pol et al.,
2010) is limited.

## 6. $^{10}$Be

The production of $^{10}$Be in the atmosphere, and its subsequent deposition to Antarctic snow, is controlled by the flux of cosmic rays, which in turn is influenced by the solar magnetic field (showing solar cycles), and on longer timescales by changes in intensity of Earth's magnetic field. As examples, centennial scale variations in $^{10}$Be in ice over the last 14 kyr can be matched to variations in $^{14}$C (Muscheler et al., 2014), while the Laschamp magnetic excursion at about 41 kyr BP (Raisbeck et al., 2017) and the Brunhes-Matuyama magnetic reversal at about 780 kyr BP (Raisbeck et al., 2006) are easily identified in ice cores. However there are, as with all aerosol-bound proxies, atmospheric transport influences on the relative amount of produced $^{10}$Be that is transported to Antarctica. Additionally, in the central East Antarctic plateau the concentration of $^{10}$Be shows a very clear imprint of climate that is mainly removed by calculating the flux. We will therefore have to independently estimate the snow accumulation rate in order to use any $^{10}$Be template for dating.

In the marine record, the strength of Earth's magnetic field is imprinted in records of geomagnetic palaeointensity. A number of reconstructions have been made using individual cores, but a carefully constructed stack from different sites is especially valuable. The PISO-1500 stack of relative palaeointensity (RPI) (Channell et al., 2009) is particularly widely used, and could serve as a template for long-term variations in ice core $^{10}$Be. In theory an even more direct comparator would be an index derived from the authigenic $^{10}$Be/$^9$Be ratios in marine sediments (Simon et al., 2018; Simon et al., 2016). Measurements extend beyond 2 Ma, and show a good correlation with the RPI (Channell et al., 2009). However because detailed $^{10}$Be data exist only for a very few cores, the RPI might be considered a more robust dataset at this stage.

Both RPI and $^{10}$Be/$^9$Be show the strong features that we know have been seen in the ice core record: in particular the Laschamp excursion and the Brunhes-Matuyama boundary. It has been reported that the PISO-1500 stack shows a good correlation with the unpublished record of $^{10}$Be flux from 200-800 ka (Cauquoin, 2013), with a correlation coefficient reported as r=0.62 after the timescales have been aligned. Unfortunately, we can only show the comparison for the few published sections of ice (Fig 8).

The extended datasets, both of palaeointensity (Channell et al., 2009) and authigenic $^{10}$Be/$^9$Be ratios (Simon et al., 2018) should therefore be useful templates with which to compare the $^{10}$Be data obtained from the LDC ice core. In Fig. 9 we show these two datasets, as an indication of what a $^{10}$Be flux record from the new core should show. Note that uncorrected $^{10}$Be/$^9$Be data automatically include the degree of decay (1.39 Myr half-life) that will also apply in the ice core, whereas the PISO1500 do not include that decay. The paleointensity lows associated with polarity reversals in particular (Fig. 9) should be quite prominent in the $^{10}$Be record (analogous to the Brunhes/Matuyama boundary). Some of the other prominent excursions events should also be captured.

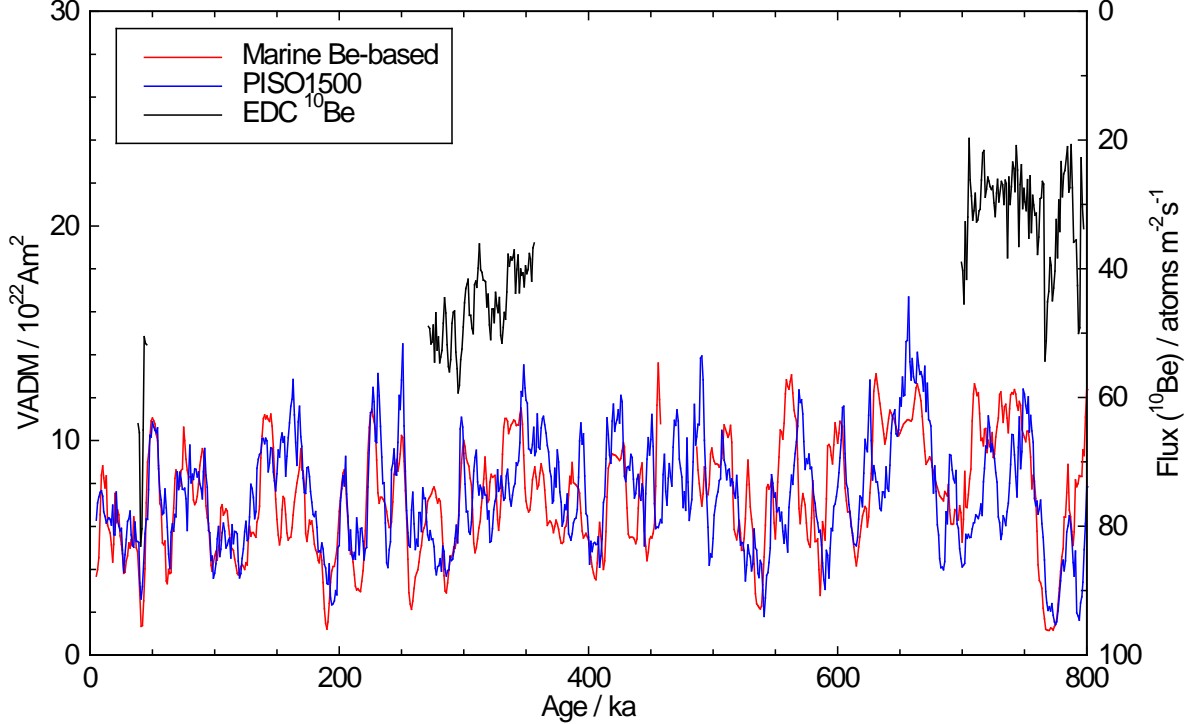


Figure 8. Palaeointensity and [10]Be data for the last 800 kyr. Virtual axial dipole moment (VADM) from the
PISO1500 palaeointensity stack (blue) (Channell et al., 2009);   VADM derived from an authigenic [10]Be/[9]Be
ratio stack (Simon et al., 2016) using an empirical calibration (red); published [10]Be fluxes from the EDC ice
core (black, right axis) (Cauquoin et al., 2015; Raisbeck et al., 2017; Raisbeck et al., 2006).

There are two aspects that degrade the ability of [10]Be alone to provide a dating template. The first is that it is the
[10]Be flux that resembles marine data, and we will only have measurements of [10]Be concentration. This issue
applies also to dust (as discussed above), but the dynamic range between glacial and interglacial for dust is so
great (factor 10 in marine sediments, higher still in ice) that the influence of accumulation rate changes is second
order and does not mask the signal that is common between ice and marine sediments. For [10]Be the range of the
data (factor 2 between low and high) is similar to the range of accumulation rates, meaning that the
concentration is equally influenced by the cosmogenic production rate and the snow accumulation rate. By itself
the [10]Be concentration will be hard to place onto the template.

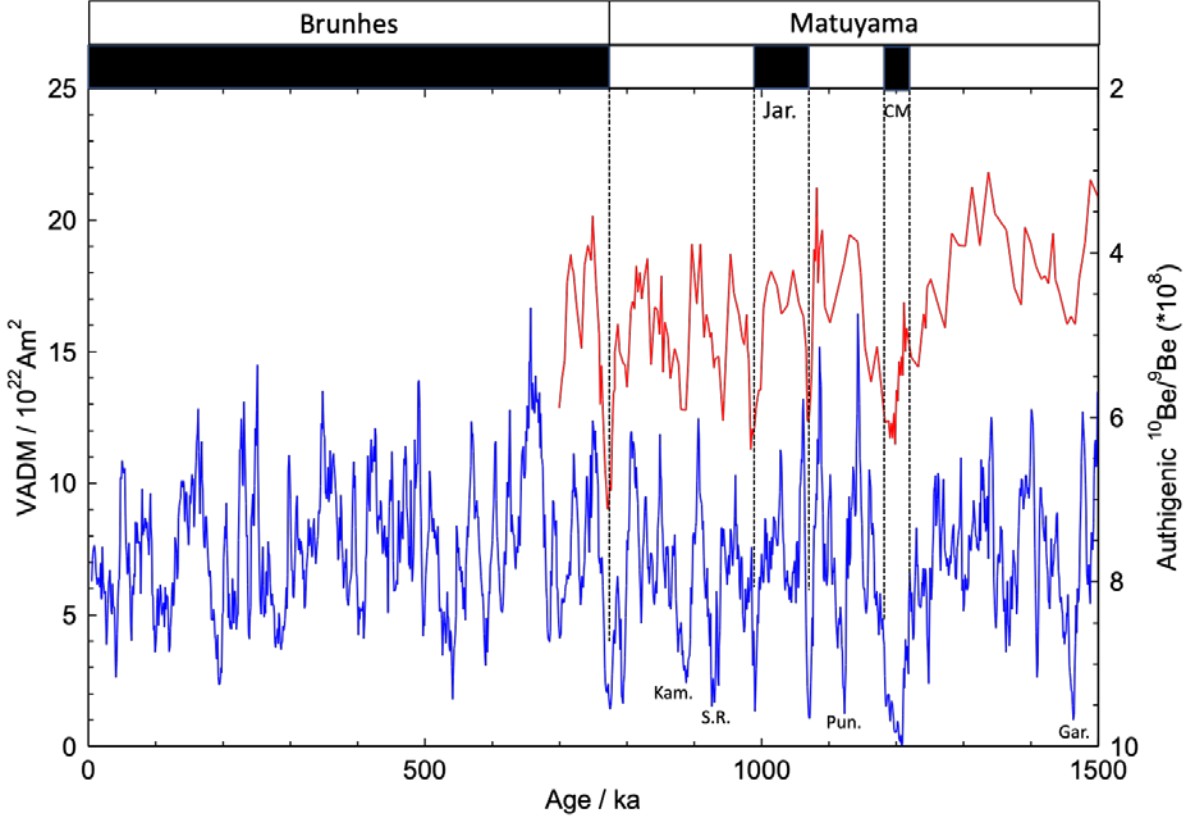


Figure 9. Palaeointensity and authigenic $^{10}$Be/$^9$Be from marine sediments for the last 1.5 Myr. Virtual axial dipole moment (VADM) from the PISO1500 palaeointensity stack (blue) (Channell et al., 2009); authigenic $^{10}$Be/$^9$Be (decay-corrected) from core MD97-2143 (red) (Simon et al., 2018). Each of the polarity reversals (Brunhes, Jaramillo, Cobb Mountain) is associated with a palaeointensity low. Other prominent excursions in the Matuyama Chron are labeled: Kam = Kamikatsura; S.R. = Santa Rosa; Pun = Punaruu; Gar = Gardar (Channell, 2017).

There is one possible way to deal with this issue. The accumulation rate for the EDC core was actually a product derived from the age modelling, but based on a prior where the accumulation rate was assumed to be directly related to the temperature and hence to the water isotope ratios. If we assume that that relationship was unchanged over 1.5 million years then the best fit values from the 800 kyr of EDC could be used, along with water isotope ratios measured at LDC to estimate the accumulation rate for each depth and therefore calculate a flux of $^{10}$Be. This will have considerable uncertainties but is likely to allow identification of the main features in the expected $^{10}$Be record.

An additional problem is the one encountered when the Brunhes-Matuyama section of the EDC ice core was analysed (Raisbeck et al., 2006), that $^{10}$Be in deeper ice shows spikes that appear to be inhomogeneous across the core and may be associated with high concentrations of dust and other chemical concentrations. The spikes have been tentatively ascribed to a concentration effect where $^{10}$Be becomes associated with dust particles which also seem to clump together into aggregates in the deeper ice (de Angelis et al., 2013). For the Brunhes-Matuyama section of the EDC ice core, the spikiness in $^{10}$Be was bypassed using median concentrations

(Raisbeck et al., 2006), and it may be that such a strategy will continue to work in older ice. However, further
work is needed to understand the conditions that lead to this effect.
**7.  Discussion and conclusion**
We have presented templates for what an undisturbed (i.e., where time is monotonic with depth) ice core from
LDC might be expected to show. We summarise the 4 methods we have considered in Table 1. The marine dust
record (represented here by Fe MAR at ODP site 1090) could, with reasonable assumptions, be an excellent
template for the LDC dust record. The Mg/Ca data from site 1123, matched against the LDC water isotope
record, could provide additional validation, although the correlation between records over the past 800 kyr is
less strong than for dust. The methane data, matched against D-O variability at site U1385 may be capable of
adding some sharper tie points in a record that has already been matched to first order. $^{10}$Be concentration,
converted to an estimated flux using water isotope data, should be a useful additional constraint, particularly in
identifying the major features with low Vertical Axial Dipole Moment (VADM) and expected high $^{10}$Be
concentration. All of the constraints provided by this method and others would be included within a Bayesian
framework using a program such as IceChrono (Parrenin et al., 2015), which would provide an estimate of the
uncertainty.

| Ice core measure | Template | Assumptions | Comments |
|---|---|---|---|
| Dust | South Atlantic marine core dust proxy | Changes in atmospheric transport do not overwhelm the common source signature | Good matches at multimillennial scale even when using ice core concentration |
| Water isotopes | Southern marine benthic Mg/Ca | Antarctic surface temperatures continue to drive southern deepwater temperature | Good match at orbital scale, but less good for cycles of weaker amplitude |
| Methane | Portuguese Margin marine planktic $\delta^{18}$O | Both records record common millennial variabiity | Millennial scale alignment possible, but different amplitudes for individual peaks would make it hard to use alone. |
| $^{10}$Be | Paleointensity measures in marine cores | Both records dominated at long timescales by strength of Earth's magnetic field | Requires estimate of ice accumulation rate to derive $^{10}$Be flux; statistical issues with $^{10}$Be spikes need to be solved. |


Table 1: Characteristics of possible template matching methods for a 1.5 million year ice core

It will be a greater challenge to use these records to aid the age modelling if the record is disturbed, with folds or
missing ice, as has often been the case with ice near the bed of ice sheets (e.g. NEEM Community Members,
2013). In that case, one cannot rely on the shape of the signal to identify the time period represented. Instead, we
are dependent on using the absolute values – for example finding a time period where the values in the templates
are all consistent with the measured values. The derived $^{10}$Be production may be particularly important in this
case, because it is independent of climate and, thus, may provide a more robust age assignment compared to the
other templates considered here, which are highly correlated on glacial/interglacial time scales. This will have to
be done with considerable caution, given the uncertainties involved in the assumptions about the unchanged
relationship between the measured values and their marine equivalents over time. Finally an age model for the
new core will of course also use other data, including those from gas measurements ($\delta^{18}O_{atm}$ and $O_2/N_2$, which
can be matched to calculated orbital targets), and any radiometric absolute ages that can be obtained from the
limited ice volumes available.
**Data availability**
All the datasets shown in this paper have already been published elsewhere, as indicated by the relevant
references.
**Author contribution**
All authors conceived the idea for this paper. EW prepared the first draft and all authors reviewed and edited the
text.
**Competing interests**
The authors declare that they have no conflict of interest.
**Acknowledgments**
This publication was generated in the frame of Beyond EPICA. The project has received funding from the
European Union's Horizon 2020 research and innovation programme under grant agreement No. 815384
(Oldest Ice Core). It is supported by national partners and funding agencies in Belgium, Denmark, France,
Germany, Italy, Norway, Sweden, Switzerland, The Netherlands and the United Kingdom. Logistic support is
mainly provided by PNRA and IPEV through the Concordia Station system. The opinions expressed and
arguments employed herein do not necessarily reflect the official views of the European Union funding agency
or other national funding bodies. This is Beyond EPICA publication number XX. This publication is also
associated with the Million Year Ice Core (MYIC) Project of the Australian Antarctic Program (AAP). We
thank Alexander Cauquoin and Grant Raisbeck for advice about the published [10]Be data. EW was supported by
a Royal Society Professorship. HF acknowledges the long-term financial support of ice core science by the
Swiss National Science Foundation. TvO acknowledges support by the Australian Government Department of
Industry, Science, Energy and Resources (grant no. ASCI000002). We thank 4 reviewers for their helpful
comments.

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
