# Peer review of "Stratigraphic templates for ice core records of the past 1.5"

_Climate of the Past, 2022_

## Author Comment (AC1)

Response to reviewer RC1

Review in italics, our response in standard text

We thank the reviewer for carefully reading our paper and for their comments.

*Wolff and colleagues present some advantages and drawbacks of available chronostratigraphic parameters to date "old ice records" (stretching back to 1.5 Ma) that should be retrieved from ongoing ice core drilling efforts in Antarctica. Providing reliable chronologies to such ice cores are of utmost importance for climatic interpretations. Dating is indeed sometimes regard as secondary compare to interpretations of elegant proxies or high-tech developments requested to recover such deep-old ices. It is however of primary importance to understand climate dynamics. In this paper, authors discuss synchronization and relative matching between ice core and sediment records using four independent ice core parameters and proxies: water isotopes vs Mg/Ca, dust vs iron acc., CH4, 10Be. The similarities already observed over the last 800 ka between ice core and marine records serve as key assumptions to propose and test matching and synchronizations of old ice records (using these proxies as analogues). Such a discussion is very welcome and this paper certainly contributes to this important goal by suggesting few templates and patterns of relative changes useful for dating. The manuscript is well written, concise and rather clear for a broad audience (not necessarily specialists of one of the presented proxies). I therefore recommend this paper for publication following minor revisions (see comments below).*

Thank you for the positive comments.

***Comments***

*Page 2, first paragraph: authors could maybe explain in few sentences the MPT conundrum and why it is particularly "hot" for understanding climate changes mechanisms.*

We are happy to do that, and will add a sentence about the nature of the MPT.

*Page 2, third and fourth paragraphs: as this paper is dealing with chronological issues, it could be interesting to read here a bit more about the dating (methods, ages, uncertainties) of the 2 Ma blue ice. For instance: what's the typical uncertainties associated with these methods? Why is it vital to reduce ages uncertainty to try understanding the underlying causes of the MPT? What is the acceptable minimum uncertainty to interpret the dynamics (amplitude, frequency) of climatic changes over the MPT? Why (and how) is it important (despite uncertainty) to adjunct radiometric ages to fix relative chronologies onto absolute timescale? Etc.*

We will add a note about the use of 40Ar to date the blue ice and quote its precision. The Yan et al paper quoted a precision for the 40Ar method of 110 kyr which is clearly insufficient for placing data within a 40 kyr climate cycle.

*Lines 139-140: please explain how you scaled up the two records. Is maximum stretching still within chronologies uncertainty? This could be interesting since it would somehow quantify the elasticity between two largely used chronologies for two important records, i.e. LR04 and AICC2012 for ODP site 1123 and EDC, respectively.*

The scaling in this sentence refers to the scaling in amplitude not in time. The two records are presented on their own age scales (AICC2012 and LR04 respectively). There are no major discrepancies between the two age models within their respective uncertainties. We will adjust the text to emphasise that it is the amplitude we scale.

*Lines 151-153: it is interesting to note that these mismatches occur within the Mid-Brunhes transition (MBT). Could it be a problem as well within the MPT where amplitude change drastically?*

Yes, this is the point we already make in line 152-3, so we don't think we need to make a further change.

*Line 199: same comment as above, explain the "appropriate scaling" a bit more (how many tie-points, maximum stretching amplitude…).*

Again it is the amplitude we have scaled. Martinez-Garcia et al (2011) carried out the graphical alignment that converted their depth scale into the ice core age scale. We simply converted their ages (on the old EDC3 age scale) into our ages on AICC2012.

*Lines 225-227: based on which arguments?*

Our argument is that the common factorial change is from the changing source strength and from changes at the start of the transport, which are common between the ice and marine records. Only if the lifetime changes completely reversed and overwhelmed the source changes would the pattern differ between the two records We would therefore expect the major features to remain in common. We will add a sentence to explain this.

*Line 269: I'm not sure whether the mention "soon to be published" without any other information is acceptable. It does not tell much to readers. It is a bit annoying since this planktonic isotope and SST records from site MD95-2042 should "serve as a regional template for D-O variability".*

To us, it seemed helpful to alert the reader that this is a record that will appear, not just a long term wish. However we do not use the unpublished data, or even hint at its content. If the editor prefers us to remove this phrase then we can do so.

*Lines 271-272: I agree, using CH4 requests testing the validity of the underlying assumption first. Is any such East Asian speleothem record already exist or is soon to be published?*

We are not aware of any longer records from East Asia. There are spelothem data over longer periods from other locations (eg Bajo et al 2020) but this does not answer the question regarding methane source regions.

*Lines 273-277: I would also cite here papers by Giaccio et al., (2015) and Nomade et al. (2019) where millennial events are also clearly identified in records that are also radiometrically date providing potentially important tie-points to fix floating chronologies.*

We thank the reviewer for bringing these interesting papers to our attention. We agree that correlations with records that have radiometric analysis could add absolute ages and we will mention this in the text.

*Line 299: add 'intensity' to "Earth's magnetic field".*

OK, we will add that.

*Lines 305-306: this is a problem since it invokes some circularity, or at least imply a strong assumption. Why not try using regression method (see Zheng et al. 2020 and 2021 for instance) to normalize old ice 10Be records and minimize climate-related variations? One could for instance use water isotopes or ion concentrations data within multi-linear correction method to obtain climate corrected 10Be record. The idea is essentially to remove the shared variance between 10Be and climatic parameters measured at the same depths. This method has been used in wet deposition environments (Greenland), but could maybe provide interesting results, even imperfects, in such dry deposition settings.*

We devote quite a lot of discussion to this issue and in lines 349-355 (original version) we suggest how the problem might be solved. It involves assumptions about the constant relationship between water isotopes and accumulation rate. Despite the assumptions, it does offer a mathematical way to remove the influence of accumulation rate which is more straightforward than looking for shared variance. We have not therefore changed the text here.

*Lines 318-319: this is very unfortunate indeed.*

Yes!!!

*Lines 323-324: authors could emphasis a bit more why 10Be radioactive decay (thus uncorrected 10Be records) could serve as an interesting indicator for long-term age control in case of disturbed or non-continuous ice cores. This was for instance use in Bourles et al. (1989) or to try dating very disturbed (discontinuous) lacustrine records (Lebatard et al. 2010; Simon et al. 2020).*

We agree that the long term decay could give a first order idea of the age, as could the other radiometric methods we mention in the introduction. The use of decay of cosmogenic isotopes for ice is mentioned in line 55, in the context of the use of the composite decay of 10Be/36Cl. Although we agree on the use of these absolute age markers in addition to relative alignments to templates, we think it could be distracting to the line of argument to discuss it again at this point in the paper.

*Lines 333-340 and 349-355: one way to try resolving this issue is to use the regression method (see above) and compare such results with estimated accumulation rate (derived from water isotopes) by an iterative approach.*

We already discuss this point under the comment for lines 305-306.

*Lines 360-363: I agree the spikiness was bypassed in EDC using statistical method. However, the 10Be measurements density in EDC allowed such treatment. Will it possible to do so within older ice samples considering the amount of material needed to measure 10Be. Further works are indeed needed and maybe 10Be measurements on drill-chips (Auer et al., 2009; Nguyen et al., 2021) will be key to obtain high-resolution 10Be records in old ice records.*

We agree that it may be difficult to correct for the spikiness in older ice. Studies are indeed underway, I believe at both Lund and at CEREGE, to try and better understand this issue. Drill chips will likely be used as well. We feel we have already highlighted the difficulty the spikiness imposes and that this is not the place where we can solve it, so we have not altered the text.

**References**

*Auer, M., et al. (2009). Earth and Planetary Science Letters, 287 (3), 453-462.*

*Bourles et al. (1989). Geochimica Cosmochimica Acta, 53 (2), 443-452.*

*Giaccio et al. (2015). Geology, 43 (7), 603-606.*

*Lebatard et al. (2010). Earth and Planetary Science Letters, 297, 57-70.*

*Nguyen et al. (2021). Results in Geochemistry, 5, 100012.*

*Nomade et al. (2019). Quaternary Science Reviews, 205, 106-125.*

*Simon et al. (2020). Quaternary Geochronology, 58, 101081.*

*Zheng et al. (2020). Earth and Planetary Science Letters, 514, 116273.*

*Zheng et al. (2021). Quaternary Science Reviews, 258, 106881.*

References:

Bajo, P., et al. (2020), Persistent influence of obliquity on ice age terminations since the Middle Pleistocene transition, Science, 367(6483), 1235, doi:10.1126/science.aaw1114.

---

## Author Comment (AC2)

Review in italics, our response in standard text

We thank the reviewer for carefully reading our paper and for their comments.

*This manuscript's goal is to provide recommendations for how best to develop age models for new ice cores that are anticipated to retrieve ice from 0.8-1.5 Myr ago. As significant resources are being invested to retrieve this ice, the thoughtful advance planning for the creation of these age models is commendable. In proposing specific age modeling strategies, the manuscript also generates useful hypotheses about the types of climate responses that are expected to be found in the new ice cores. This manuscript is well written and appropriately aimed at a relatively broad paleoclimate audience. I recommend publication after minor revision.*

Thank you.

*My suggestions for revision are below:*

1. *Most importantly, it would be useful for the introduction to be more explicit about the practical goals for dating the new ice. Although the manuscript currently introduces the motivation to compare pre-MPT and post-MPT climate responses, it would be useful to list a couple specific hypotheses to be tested that will require good age models and how precise the age models will need to be to evaluate these hypotheses. For example, how well do ages need to be constrained to evaluate shifts in orbital-scale spectral power across the MPT? Do the authors also hope to be able to evaluate leads and lags in the climate system? If so, how will the proposed alignments to sediment core records affect the ability to measure the relative timing of different climate responses?*

We will add two substantial sections discussing the reasons for seeking precise ages, and the fact that using marine records precludes seeking leads and lags between marine and ice records.

2. *At the end of the manuscript, it would also be useful to propose some criteria that might be used to evaluate the success or uncertainty of the proposed age models.*

We agree it will be important to put uncertainties on the ice age scale that is derived. This will come from a number of constraints, likely within a Bayesian framework such as that used by the Icechrono and Paleochrono programs. We have added a sentence near the end to discuss this.

3. *Line 75: Please clarify that Prob-Stack (Ahn et al., 2017) does not have its own age model. Because Prob-Stack uses the LR04 stack as its initial alignment target, it has implicitly inherited the LR04 age model and, thus, has at least as much age uncertainty as the LR04 stack.*

Thank you for the clarification, which we will add to our text.

4. *Line 108: This sentence is unclear. What exactly is "higher" at Dome Fuji?*

Concentration of dust is higher at Dome Fuji – this will be clarified in the text.

5. *Lines 151-153: Please describe the characteristics of the deep water temperature signal from 450-550 ka that are also found in some of the older sections of the 1123 signal. Are there specific ways to identify in advance which parts of the 1123 signal may be problematic for alignment?*

We were referring to the similar amplitude of variations. This will be spelt out in the new version.

---

## Author Comment (AC3)

Response to reviewer RC3

Review in italics, our response in standard text

We thank the reviewer for carefully reading our paper and for their comments.

*Wolff et al. address the challenging problem of dating ancient ice from Antarctica with ages exceeding 800 ka BP – currently the oldest ice in the continuous ice core record. Such dating is challenging in particular when the ice is no longer in stratigraphic order. The authors suggest 4 ice core records (dD, dust, CH4 and 10Be) that may in theory allow for stratigraphic matching to independently-dated marine-sediment records (of benthic Mg/Ca, Iron, Atlantic SST and geomagnetic dipole, respectively).*

Thank you.

*The paper is well-written and easy to follow. The topic is suitable for Climate of the Past. I have a series of comments for the authors to consider, in particular for an improved discussion of the approach.*

*(1) I would ask the authors to provide a more in-depth comparison of the methods, including perhaps a ranking of the methods. There is currently no quantitative analysis of the four methods they propose – this could be remedied by adding a table with the correlation coefficients between the ice core record and the marine target record for each of the methods proposed.*

*Such a table could also include an assessment of how likely the physics that produce the correlation for the last 800kyr is to persist during 0.8 – 1.5 M also. Such an assessment is likely qualitative and somewhat subjective by necessity (e.g. "likely", "unlikely", "uncertain" etc).*

*A third entry into the table could for example be whether the method would allow for value-matching, or only the matching of sequences.*

*Combined, these elements would allow the reader to assess the relative usefulness of each method. The authors appear to give somewhat of a ranking of the four methods starting on line 366. The authors appear to favor the Dust-Fe matching over the dD-Mg/Ca matching – a choice that I agree with. This ranking is however never stated explicitly, or justified by the analyses.*

We will provide, as requested, a correlation coefficient for the isotope and dust matches, where the whole record is involved as a template. This cannot be done for the 10Be where we have only a few snapshot periods of comparison at present. It also does not make sense for methane, where we point out that the strength of millennial scale events is different between methane and either water isotopes in ice or isotopic data in marine sediments. We point out that the utility of the methane method would be to provide millennial scale interpolation between records that have already been synchronised by other methods.

For information, Elderfield et al gave a correlation coefficient of 0.67 (r^2 = 0.45) for 1123 Mg/Ca temperature against ice core delta-D based temperature (after alignment). For ice core

dust against site 1090 Fe MAR, we calculate a correlation coefficient of 0.83 (r^2 = 0.69). This is the same for dust flux and dust concentration in the ice (see response to reviewer 4).

We have considered providing a table showing the characteristics of each method. However we believe that the textual description we give in the first paragraph of section 7 provides the information requested, and that a table would be an inflexible way to do the same job. We do not favour a ranking as we believe that it will be most advantageous to use as many methods as possible, to ensure that false matches are not made, and to take advantage of the strengths of each method.

*(2) I have reservations about the dD to Mg/Ca matching. To me, one of the most exciting prospects of a 1.5 Ma ice core record would be to investigate the Antarctic dD climate record and its spectral properties. This possibility would be lost if it were wiggle-matched to the deep ocean temperature.*

*A surprising aspect of the 41-ka world is the absence of a precession (21 ka) signal in the benthic d18O. The hypothesis by Raymo et al. (2006), paper cited in the text, is that the NH and SH ice volume each responded to local precession forcing, yet this cancels out in the benthic d18O (and presumably also in benthic Mg/Ca) as the precession forcing is out of phase between the hemispheres. One of the key questions of a 1.5Ma ice core dD record is whether it has power in the precession band in the 41 ka world – the Raymo hypothesis requires that it does. If the dD record is wiggle matched to a benthic record, we would lose the ability to independently assess the differences in spectral content between such records – one of the key scientific objectives.*

*The authors suggest that the ODP 1123 benthic Mg/Ca resembles Antarctic dD so strongly because both are controlled by SH high-latitude SST. They follow the argument by Elderfield (2012) here (Lines 125-129). Antarctic dD also strongly resembles global benthic d18O (LR04), and it also strongly correlates with mean ocean temperature from ice cores (Shackleton et al., 2021; Fig. 3). I suspect that LR04 may actually give a better correlation to dD than ODP1123 Mg/Ca does (can you check/show?). Since bottom waters formed around Antarctica are at the freezing point, lowering Southern Ocean SST will not cool them further. It seems to me that global mean ocean temperature, the rate of bottom water formation, and circulation may all impact ODP1123 benthic Mg/Ca. ODP1123 certainly looks identical to LR04 (Elderfield 2012) in d18O, suggesting good connectedness to global ocean conditions.*

*For these reasons I would ask the authors to reconsider recommending wigglematching dD to ODP1123 as a dating strategy.*

We fully understand the reluctance to use matching between marine and ice records because of the fact that this removes any possibility to investigate phasing between records. We would certainly favour methods where there is an a priori reason to expect the common signal to be recorded simultaneously in ice and marine records. This is the case for cosmogenic isotopes with their common production signal. It is partly the case for dust (common production signal) but we can expect some additional influence from overlying transport/lifetime signals that appear to reinforce the production-based signal, but cannot be assumed to be perfectly in phase. It's correct that it is a less strong constraint for a temperature signal, although the mechanism proposed by Elderfield et al (2018) would lead to a similar timing of the temperature signal in ice and deepwater. We will certainly adjust the text to make sure this point is understood.

However, it is important not to forget the underlying reason why we thought this paper was necessary: that it may be quite challenging to date the ice by other means. We would certainly prefer not to use correlations with records (orbital or marine) that require phasing assumptions that cannot then be tested. But without such methods we may have no age scale at all, which certainly precludes the phasing tests discussed by the reviewer.

We are less convinced by the suggestion to compare ice core water isotope records directly with the benthic isotope record from LR04. That record is indeed well-correlated with the EDC isotope record, but it is a mixed signal of deepwater isotopic content (hence ice volume) and deepwater temperature. While it's true that everything looks similar on glacial-interglacial timescales, there are good reasons to expect a more robust mechanistic link between the directly measured Mg/Ca deepwater temperature proxy and the ice core temperature proxy than between the mixed benthic signal and the ice core proxy. Probably the ice core mean ocean temperature signal would be an even better comparator, but it will be a while before a continuous high resolution record of that is available.

*(3) could you elaborate on the temporal resolution needed to do the CH4 / planktic d18O matching, and whether this resolution is available in the planktic record (which is promised but unfortunately not shown). For example, planktic d18O misses a DO event around 778ka BP that is clear in the EDC CH4. Could there be DO events that are missed altogether by both records?*

The resolution of the planktic record shown in Fig. 7 is about 200 years. The resolution of the methane record in the deepest part of EDC is typically about 500 years. This is sufficient to resolve millennial scale features, though not some of the sharper D-O events that are seen in the last glacial cycle. We discuss problems of resolution that might arise in older ice (lines 291-295 of original paper). We will add a sentence about resolution of the planktic record.

*(4) if Raymo et al. (2006) is right, the MPT reflects a transition from terrestrial to marine terminating Antarctic ice sheets. If so, could we get local Antarctic dust sources to contribute to the ice core record? Could this impact the dust matching?*

This is an interesting point. If the ice retreated significantly, uncovering large areas of new dust sources it could indeed influence the dust record. We thank the reviewer for reminding us of this, and have added a further caveat to the text. In particular we mention the importance of checking for new dust sources through isotopic fingerprinting such as that used by Delmonte et al (2008).

*(5) line-by-line comments:*

*Line 40-44: acknowledge the Japanese and US ice core communities are also working on this in funded projects.*

We already write (line 40 of original text) "Several projects to obtain such a core are partially underway". We then specify the two funded projects in the Dome C region as it is the Dome C record we use as the ice core template. We are a little reluctant to start naming every nation/group with a plan, as this would also include Chinese, Korean, and Russian groups as well as Japanese and US. These plans are at various stages of readiness, and we do not have the information to confirm which of them actually have the funding to drill a deep ice core (for example the US Coldex project is funded for oldest ice work but it would not be true to

say they are funded to drill a deep ice core). Rather than getting ourselves into a diplomatic quagmire, we prefer to refer to "several projects" and then specify the two we know best and which take place in the region from which we have taken ice core data.

*Line 65: other tuning targets are air content and d18O-O2.*

Indeed. We used the word "including" to indicate a non-exhaustive list. Since we do mention d18O-O2 in the next paragraph we will now include that (and a reference) in this paragraph.

*Line 115: I think both cited studies suggest dD is a proxy for site temperature – they just disagree as to what the correct calibration is.*

But the reason why calibrations are uncertain is because water isotopes are influenced by many factors and processes not all of which are directly related to temperature. We are happy with our wording here.

*Line 128: The temperature of the deep waters formed around Antarctica is probably always close to the freezing point. Could it not reflect the volume of deepwater formation, for example, and the mixture with other deep water masses?*

While no doubt there are many factors involved in deepwater formation, our statement correctly characterises what the Elderfield paper hypothesised. In fact if you accept their Mg/Ca dataset as representing the deepwater temperature, then it is not always at the freezing point, but actually varies over a range of about 4 degrees. We have not made a change here.

*Line 131: You could add Shackleton 2021 here too (new MOT data from MIS 4). In that paper she explicitly plots the very strong correlation between MOT and ice core d18O, which strengthens your argument as you correctly note.*

OK, reference added.

*Figure 3: is it possible to also plot the comparison as a Mg/Ca vs. dD scatter plot? That way the reader can assess the correlation better.*

*Line 146: can you give the correlation coefficient for the comparison?*

We will provide a correlation coefficient (value given in response above). A scatter plot does not seem necessary to us in addition.

*Line 177-180: I fully agree with this sentiment, but feel the same logic can (should?) be applied to ODP 1123. The Benthic temperature has an imprint of southern high-latitude SST, but also of global ocean temperature, volume of deep water formation, circulation, etc.*

I think we have to agree to differ here. We agree that the close connection of Antarctic temperature to deepwater temperature at a southern site cannot be guaranteed to be simple or static and we discuss this in some detail (lines 154-166 of original text). However, it is certainly much more closely connected to Antarctic temperature than a record of water isotopes in the North Atlantic, which contains two controlling components: one (ice volume) dominated by northern hemisphere ice, and the other (deepwater temperature) now at a site

where both northern and southern waters set the temperature. We agree that we could derive a good observational relationship between deuterium in ice over the last 800 kyr and the benthic record (whether from the Portuguese Margin or from the LR04 stack). However given the multiple controls on benthic isotopes we feel that the argument for constancy of the relationship across the MPT is much weaker than for the ODP1123-Antarctic temperature relationship. Our discussion around this point will allow the reader to make their own judgement whether they agree with us.

*Line 203: How good is the match? Can you quantify, e.g. via a correlation coefficient?*

We will include a correlation coefficient for the dust match (value given in earlier response above).

*Line 255: Perahaps note that DO 2 has no CH4 peak at all.*

We think this is too detailed as it would require us to number DO events (so readers know what we mean) and then discuss an event that is actually quite hard to pick even in the NGRIP water isotope record at the scale used here.

*Line 268: Throughout what? Throughout the record? throughout the Pleistocene?*

We have added "throughout the past 1.45 Myr".

*Line 269: can you give more details than "soon to be published". Is there a title and author list?*

To us, it seemed helpful to alert the reader that this is a record that will appear, not just a long term wish. However we do not use the unpublished data, or even hint at its content, and we do not believe it's appropriate to list it as a "paper in preparation" with full bibliographic details. If the editor prefers us to remove this phrase then we can do so.

*Line 317-318: This is unfortunate (though no fault of the authors). why is the full dataset from a 2013 publication not publicly available at this point? Can you provide more details?*

As the reviewer will note the data are shown in a thesis but have never been fully published, hence the lack of a dataset. We cannot answer the question posed.

*Line 366: Here the authors appear to suggest a hierarchy with the dust matching being considered more accurate than the other methods. Is this really what you mean to imply?*

As discussed above, we see the different records as complementary, each with their own strengths. Dust could indeed be superior to water isotopes but not if (as the reviewer pointed out) we find new dust sources for example. Hopefully the text in section 7 clarifies the strengths and weaknesses of the methods.

*Line 378: However, wouldn't the absolute values of the 10Be be challenging to use due to the accumulation uncertainty? Also, the VADM record is really variable, and certain 10Be values are not necessarily very unique.*

Yes. We wanted to inject some optimism as a lot of effort will go into producing the 10Be record. However we don't want to minimise the issues with using it in practice and hope we reflect this balance in our text.

---

## Author Comment (AC4)

Response to reviewer RC4

Review in italics, our response in standard text

We thank the reviewer for carefully reading our paper and for their comments.

*In anticipation to the new ice core data beyond the Myr that the community will generate, Wolff et al. propose an overview of the possibilities that the community could have to correlate and date such old records. As per request from the handling editor, because the other reviewers have already commented on other points of the MS, and because this is indeed my expertise, I have concentrated my review only on the dust part of the paper.*

*I have two major issues :*

*- The authors compare the dust flux in ice on one hand, and an iron concentration on the other, justifying it by the fact that claim dust flux is what they will get in older ice cores, and claiming it is fair to compare it to concentration. I am sorry, but to me it is a bit like comparing apples and pears. And it has been a redundant problem within the dust community working on different archives. As the authors mention, to calculate a flux, one needs ages and accumulation rates, but those accumulation rates would also influence a concentration profile when using it. It is commonly accepted that fluxes or at worse ratios should be priviledged when comparing dust proxies between archives, so it is accept that fact and how to solve this issue if this is problematic for some archives or time intervals. Otherwise, again, it makes any comparison quite speculative despite a match which is indeed surprisingly good.*

Some of the authors of this paper also have considerable expertise in dust (in ice), and have indeed written extensively about wet and dry deposition, and about the importance of considering either fluxes or concentrations under various circumstances in ice core studies. In this case we fully agree that flux would be the better comparator to the marine record. But this wish doesn't alter the fact that we do not measure flux, we measure concentration. To derive flux in an ice core requires a set of assumptions that may or may not be correct as we go further back in time. Indeed we discuss doing exactly that in the case of 10Be. In the case of dust we simply note that we are lucky and the concentration already shows a great correlation with the marine dust flux. It turns out that the r^2 value for a correlation of log (ice core dust) vs log (marine flux) is 0.69 both for the ice core flux and the ice core concentration. This is (we suggest) because the dynamic range of the ice core dust concentration (factor 100) is so great that the changes in accumulation rate (factor 3) are not very important. This is not the case in the marine record where the dynamic range is smaller. Ratios are used to normalise for flux in the marine record; this is not an option in the ice core record where there is no marker expected to have a constant flux. We have spelt this out in the revised text. We will also show the flux in Figure 5 as an additional comparison.

*- What is "appropriate scaling". This is somehow linked to my previous comment as it feels like two curves which are in theory not comparable, can be perfectly "matched". There is no details whatsoever how this scaling is achieved. This is also valid for other matching in the MS, and also more generaly in the literature matching ice and marine records. There is absolutely no details on how it is achieved, and it would be good to provide a detailed explanation on this, including the calculations, how it is done ("stretching-compressing" method?, but how, on which time interval, varying depending on the time interval, etc?), all*

*that in supplementary data, so everybody could understand and more importandly, reproduce it.*

There may be some misunderstanding here. We have simply scaled the y-axis of the two records so that peaks and troughs in the two records match. They are both on log scales in Fig. 5, so no additional vertical stretching is involved. For the time axis we used the matching already carried out by Martinez-Garcia et al (2011). Such matching is typically done using a program such as Match (Lisiecki and Lisiecki 2002) which uses dynamic programming to find the optimal alignment of two paleoclimate signals. We will clarify this in the revision

*Minor comments/questions:*

*- I am ot a specialist in this, but really no means to date both marine and ice cores beyond 1Myr? That would help generating fluxes... And perhaps this is something the community should concentrate on?*

We agree that improved and independent means of dating are important and note this comment. This is certainly something the community is focussed on and indeed is central to prompting us to write this paper.

*- I think the reference citing dust from Patagonia to Antarctic could be reviewed by adding more recent references than only Delmonte, 2008.*

While there are more recent papers looking at dust provenance in other cores and over particular time periods, this 2008 paper remains the most comprehensive effort at dust provenance over the 800 kyr period.

---

## Author Response (AR2)

Author response for "Stratigraphic templates for ice core records of the past 1.5 million years" by Wolff et al.

The only editorial comment to the last version was to avoid referring to an unpublished paper. We have therefore changed the relevant sentence to: "Thus planktonic isotope and SST records from that site could serve as a regional template for D-O variability", which avoids any suggestion of unpublished data but rather urges that such data should be obtained.